# Preparation of a Micronutrient-Enriched Apricot Kernel Oil and Assessment of In Vitro Chemopreventive Properties [note 1]

**DOI:** 10.3390/ijms26189237

**Published:** 2025-09-22

**Authors:** Melania Elettra Vaccari, Valeria Cavalloro, Martina Bedeschi, Patrizia Serra, Giorgia Simonetti, Emanuele Casali, Alessio Porta, Alice Fossati, Emanuela Martino, Simona Collina, Anna Tesei

**Affiliations:** 1IRCCS Istituto Romagnolo per lo Studio dei Tumori (IRST) “Dino Amadori”, 47014 Meldola, Italy; melania.vaccari@irst.emr.it (M.E.V.); patrizia.serra@irst.emr.it (P.S.); giorgia.simonetti@irst.emr.it (G.S.); anna.tesei@irst.emr.it (A.T.); 2Department of Earth and Environmental Sciences, University of Pavia, 27100 Pavia, Italy; valeria.cavalloro@unipv.it (V.C.); alice.fossati01@universitadipavia.it (A.F.); emanuela.martino@unipv.it (E.M.); 3NBFC-National Biodiversity Future Center, 90133 Palermo, Italy; 4Department of Chemistry and INSTM, University of Pavia, 27100 Pavia, Italy; emanuele.casali@unipv.it (E.C.); alessio.porta@unipv.it (A.P.); 5Department of Drug Science, University of Pavia, 27100 Pavia, Italy; simona.collina@unipv.it

**Keywords:** apricot kernel oil, antiproliferative activity, colorectal cancer, nutraceutical ingredient

## Abstract

Apricot kernels (*Prunus armeniaca* L.) represent a valuable by-product of stone fruit cultivation, offering diverse applications in food, cosmetic, and pharmaceutical industries. While apricot kernel oil is recognized for its rich composition of unsaturated fatty acids, phenolics, and tocopherols, its therapeutic potential, particularly in cancer prevention, remains unexplored. This study investigated a purified fraction (FOPF) obtained from Farclo variety kernel oil, cultivated in the Emilia-Romagna region of Italy and selected for its naturally low amygdalin content. In vitro studies demonstrated FOPF’s significant antiproliferative effects against colorectal cancer (LoVo, HT29) and hepatocarcinoma (Hep3B) cell lines, with GI_50_ values ranging from 0.06 to 0.09 mg/mL. The fraction induced cell cycle arrest and significantly inhibited cancer cell migration, effects mediated through PPAR-γ expression modulation. These findings establish FOPF’s potential as a natural chemopreventive agent and provide a foundation for its development as a nutraceutical ingredient targeting colorectal and hepatic cancers.

## 1. Introduction

Apricot (*Prunus armeniaca* L.) cultivation ranks as the third most economically significant stone fruit, valued for its high concentration of essential micronutrients, important for human nutrition. This fruit, rich in polyphenols, vitamins, and carotenoids, can be eaten fresh, dried, or processed into juices and canned products [1]. The kernels, also known as bitter almonds, are commonly discarded and represent waste products during apricot processing. This is due to the presence of the cyanogenic glycoside amygdalin, which is responsible for the bitterness and potential side effects, as its in vivo metabolism produces hydrogen cyanide (HCN) [2]. The apricot oil is obtained from kernels. Constituted of approximately 50% of unsaturated fatty acids, 23–27% of protein content, and 32–34% of essential amino acids [3], it is enriched with total tocopherol [3], and possesses a high mineral content, particularly potassium, and magnesium [4]. The yield and composition of the oil show significant variation, influenced by factors such as geographical origin, apricot cultivar, and extraction method. The amygdalin concentration also varies with cultivar, field altitude, and the state of maturation and allows us to classify apricots as sweet (HCN content between 30.2 and 79.2 mg/100 g) and bitter (HCN content between 220.55 and 317.7 mg/100 g) [5]. A crucial role in defining oil composition is also played by the applied extraction method. Conventional mechanical or chemical oil extractions can lead to thermal degradation, oxidation, and oil losses [6]. Novel extraction methods (ultrasound or microwave-assisted extraction, supercritical fluid extraction, and enzyme-assisted extraction) offer several advantages, such as potentially improving yield, preserving oil quality, and promoting environmentally friendly operations [3]. Regarding safety, the main concern with apricot kernel extracts is the presence of amygdalin, which is metabolized in vivo to hydrogen cyanide (HCN) and, at high exposure, may cause severe adverse effects. In addition, clinical trials of laetrile/amygdalin have historically failed to demonstrate therapeutic benefits, while consistently raising concerns about cyanide-related toxicity [7]. Based on this evidence, any translational application must carefully balance potential efficacy against safety, particularly considering that high exposure may result in severe side effects, with a reported lethal dose of 0.5–3.5 mg/kg in adults. To mitigate this risk, various processing methods can be employed to reduce amygdalin content in kernel oil, including grinding and soaking, as well as more advanced approaches such as fermentation, enzymatic treatment, ultrasound, or microwave processing [5,8].

Apricot kernel oil is widely used in the cosmetic industry and is also being investigated for pharmaceutical applications [9]. Its ability to fight the burden of microbial infections has been demonstrated, acting as an antibiofilm agent, being able to inhibit biofilm formation, and acting against mature biofilm of five different pathogens [10]. Other health benefits associated with the apricot kernel are anti-inflammatory and antioxidant activities [11]. Moreover, it has been evidenced that apricot seed oil contains metabolites with chemopreventive and antitumorigenic properties, making it a promising candidate for exploration as a functional food ingredient [2,12]. For example, good results have already been achieved in the chemoprevention of colon cancer and tongue squamous cell carcinoma after cell-based assays [13,14].

This study characterizes the kernel oil from the late-ripening, high-yield *Prunus armeniaca* Farclo variety, predominantly cultivated in northern Italy [15], distinguished by its low amygdalin content and superior organoleptic properties. The purified extract, enriched with gamma-sitosterol, oleic acid, and its derivative (Z)-9-Octadecenoic Acid-2,3-dihydroxypropyl ester, exhibits promising anticancer potential, as these compounds have been shown in previous studies to induce cell cycle arrest [16], promote reversion of cancer cells to healthy phenotypes [17], and exert broad anticancer activity [18]. This research further investigates its potential as a chemopreventive nutraceutical to reduce the risk of colorectal cancer (CRC) and hepatocarcinoma (HCC), two malignancies often associated with lifestyle and metabolic factors.

## 2. Results

### 2.1. Apricot Kernel Oil Purification

The apricot oil furnished by Società Agricola Guidi di Roncofreddo s.s. was obtained by the cold press of seeds derived from the production campaign of 2018. The first step of the present work was to assess the amygdalin content of the oil to evaluate its safety. Results highlighted that the concentration of the considered metabolite was 10 mg/kg of oil, which is a value below the one obtained with other cultivars, but still too high to be considered safe. Indeed, using the same extraction procedure, amygdalin content can reach 400 mg/kg of oil [19], while the concentration considered as safe is below 5 mg/kg of oil [20]. Consistently, further purification of the crude oil was necessary to develop an extract with a good safety profile. To achieve this aim, the selected fractionating methodology was mw-heated liquid–liquid extraction, this heating being able to increase the partitioning of metabolites for many solvent systems. This procedure was pursued to achieve two different goals: lowering both amygdalin content and the lipophilic fraction, thus reducing toxicity of the extract and enhancing its handling. In detail, microwave induction can cause the degradation of amygdalin, drastically reducing its content [6]. The apricot seed oil was then suspended in ethanol, and the biphasic system was subjected to three cycles of mw heating. The ethanolic phase evaporated under reduced pressure. The analysis of the newly obtained fraction was performed by gas chromatography. To achieve this aim, an extensive GC/(EI)MS (70 eV source) analysis equipped with a single quad analyzer was performed. The instrumental original datasets were analyzed with two different softwares: The NIST Mass Spectra Search Program (https://chemdata.nist.gov, version 2.4, accessed on 25 March 2020) and AMDIS v 2.73. For library search we used a built-in library search by NIST using the 2020 library. The resulting structure proposed by this software with an absolute probability gave us the structures of more abundant metabolites. In particular, we plotted not only the relative abundance of the selected substances but also the library matching factor. Results evidenced the presence of oleic acid (27%), (Z)-9-Octadecenoic Acid-2,3-dihydroxypropyl ester (15%), gamma-sitosterol (32%), endowed with antidiabetic and antihyperlipidemic properties [21], squalene (6%) a triterpene known for its antimicrobial [22,23] and anticancer properties, and stigmasta-diene (4%). Owing to the high percentage of these two metabolites, the composition in micronutrients cannot be determined. Consistently, the fraction obtained was re-dissolved in methanol and extracted again three times with hexane. The methanolic fraction was evaporated at a reduced pressure, furnishing a white solid, by now called Farclo purified fraction (FOPF). FOPF was analyzed using the previously developed GC/(EI)MS method. The outcomes of this analysis allowed us to identify the most probable structures, also according to the metabolic pathways of the analyzed biomass (Figure 1).

Results highlighted the presence of residual fatty acids, i.e., ricinoleic acid (1.06%), oleic acid (16.72%), and (Z)-9-octadecenoic acid-2,3-dihydroxypropyl ester (21.16%). The other major metabolites are sucrose (6.51%), stilbene-derivative (6.22%), gamma-Tocopherol (5.87%), and gamma-sitosterol (26.21%). The double bond arrangements, for gamma-sitosterol in particular, were determined with an extensive analysis of fragments obtained in ion-source of an EI instrument.

### 2.2. Purified Apricot Seed Oil Inhibits In Vitro Cancer Cell Growth

We employed a panel of gastrointestinal cancer cell lines, including two colon cancer cell lines (LoVo and HT29) differing in their DNA repair capacity (MMR being deficient in LoVo), and the hepatocellular carcinoma (HCC) cell line Hep3B. These cell lines are widely used in drug metabolism and cytotoxicity assays [24,25,26,27]. The effect of FOPF on cell growth was determined after 72 h of daily repeated administration, using 0.001 mg/mL, 0.01 mg/mL, and 0.1 mg/mL concentrations. As shown in Figure 2A, FOPF showed a significant cytostatic effect at the maximum concentration tested, inducing cell growth arrest in all the tested cell lines with a 50% growth inhibition (GI_50_) ranging from 0.06 mg/mL in HT29 cells to 0.09 mg/mL in Hep3B cells. Cell growth arrest, as evidenced through microscopy, was also accompanied by a considerable detachment of viable cells from the surface of the flask, compatible with anoikis (Figure 2B). Consistent with anoikis, we also detected a decrease in PPAR-γ expression level in the tested cell lines after 72 h of the treatment with FOPF, as previously found [28] (Figure 2C). In particular, the gene expression analysis showed that PPAR-γ expression level was significantly decreased in respect to untreated cells either in Hep3B (*p* < 0.0001) and in LoVo cells lines (*p* < 0.0001). In HT29 cells after treatment with FOPF the expression of PPAR-γ is not valuable.

Furthermore, flow cytometric analysis demonstrated cell cycle alterations compatible with cell growth inhibition in all cell lines investigated. As shown in Figure 3A, LoVo and Hep3B treated cells displayed a significant accumulation of cells in G0–G1 phase in respect to the untreated cells (increase of 5% in Hep3B cells and 13% in LoVo cells). In addition, LoVo cells also showed a significant decrease (50%) in cells in S phase. Regarding Hep3B cells, they also exhibited a 25% decrease in G2-M phase compared to the control. Conversely, HT29 cells did not exhibit a significant accumulation of cells similar to that observed in LoVo and Hep3B cells. However, after 72 h of FOPF treatment a significant percentage (15%) of sub-G0/G1 population was observed, suggesting an induction of programmed cell death. Afterwards, Annexin V/FITC test was performed in all three cell lines. Results showed an absence of apoptosis and necrosis in LoVo and Hep3B treated cells (Figure 3B), confirming that FOPF exhibits only a cytostatic effect but no cell killing activity in these two cell lines, as emerged in MTS assay. Furthermore, as expected based on cell cycle analysis, apoptotic cells were detected either in the early (16%) and late (10%) apoptosis phase in HT29 cell line.

### 2.3. Purified Apricot Seed Oil Hampers Cancer Cell Migration Rate

Wound-healing assay was conducted to assess the impact of FOPF on the migratory capability of HT29 and LoVo cells, both of which display varying degrees of hepatic metastatic potential [29]. FOPF was used at the concentration of 0.001 mg/mL, 0.01 mg/mL, and 0.1 mg/mL. After 72 h of exposure with repeated administration, LoVo and HT29 cells lost their ability to migrate. The migration inhibition was observed with a higher concentration in LoVo cells (Figure 4A). In HT29 cells, the highest concentration tested determined a complete detachment of cells and, for this reason, the effect on invasion blockage was observed at a lower concentration of 0.01 mg/mL, as shown in Figure 4A. In particular, in LoVo cells treated with FOPF, the percentage of cell-free area, calculated using ImageJ (version 1.54f, Wayne Rasband and contributors National Institute of Health, USA), was significantly higher compared to untreated cells (63% vs. 37.3%, *p* < 0.009) as early as 48 h (Figure 4B). After 72 h of exposure, the cell-free surface area was still significantly higher compared to control cells (49.6% vs. 2.55%, *p* < 0.0001). In HT29 cells, characterized by a lower growth rate compared to LoVo cells, the difference in gap area between control and treated cells was significant only after 72 h (58.7% vs. 40.4%, *p* > 0.003).

## 3. Discussion

The apricot seed oil obtained by the cold press was the starting point for the development of a new pharmaceutical ingredient with antitumoral potential. Preliminary results highlighted that the amygdalin content of the oil was over the limit generally considered as safe, so further purification was necessary to decrease its concentration before evaluating the anticancer potential in vitro. In detail, a purified fraction was prepared, applying a microwave-assisted liquid/liquid extraction at 60 °C. Microwaves are widely used for natural product extraction, being more effective than traditional extraction techniques and bringing a series of advantages to the table: low solvent consumption, short extraction times, and higher efficiency [30,31]. Specifically, microwaves are also able to reduce the content of amygdalin by degradation. Two subsequent mw-assisted extractions were performed, using oil/ethanol and then methanol/hexane as biphasic systems. At the end of the first part of the process, the fraction appeared as an oil. Its composition was primarily characterized by the presence of gamma-sitosterol and oleic acid, which collectively represented over a half of the total phytocomplex presenting a dual challenge. First of all, the high lipophilicity of the mixture can cause issues in the subsequent biological assays due to poor solubility, necessitating the use of a high percentage of dimethylsulphoxide to solubilize. Moreover, the high percentage of these metabolites could dilute the effect of micronutrients present in the fraction. For these reasons, we performed a second liquid/liquid extraction obtaining a purified fraction enriched in micronutrients, named FOPF. The extraction process is summarized in Figure 5.

Its chromatographic fingerprint was drawn with a GC/(EI)MS analysis. This methodology allows us to both discriminate among different chemotypes and to perform a rigorous characterization of a complex matrix. Results highlighted that the residual fatty acids represented about 39% of FOPF, while the other major metabolites were sucrose (6.51%), stilbene-derivative (6.22%), γ-Tocopherol (5.87%), and γ-sitosterol (26.21%). These compounds are of particular interest, being already associated with activities related to cancer prevention and therapy [19]. The residual cyanogenic compound present in FOPF was 2-methyl-benzonitrile, although it constituted only 1.5% of the fraction, thus not representing a serious threat for safety. FOPF’s chemopreventive properties were subjected to an in-depth biological investigation across three distinct cancer cell lines: LoVo (colorectal cancer), Hep3B (liver cancer), and HT29 (colorectal adenocarcinoma), widely utilized in drug metabolism and cytotoxicity assays.

The observed inhibition of cell growth in all three cell lines is hypothesized to be a result of induced anoikis, as evidenced by morphological features such as a floating phenotype and reduced cell size compared to control cells. Additionally, the significant decrease in PPAR-γ gene expression levels in all three cell lines treated with FOPF further supports the anoikis hypothesis. The effect was particularly pronounced in HT29 cells, where PPAR-γ expression was completely abolished after the treatment, and in Hep3B cells where it was reduced by 50%. These biomolecular characteristics of anoikis align with previous findings by Terasaki et al. in DLD-1 human colorectal cancer cells treated with fucoxanthinol and by Schafer et al. in Hep3B cells using PPAR-γ inhibitors [27,32]. Consistent with this evidence, FOPF induced substantial alterations in the cell cycle in all three cell lines tested. LoVo-treated cells exhibited a dramatic reduction in their S phase reservoir, while Hep3B treated cells displayed an altered cell cycle profile compared to the control, with a significant reduction in the G2-M phase. HT29 treated cells exhibited a significant population of cells in the sub-G0–G1 phase, suggesting an inability to undergo cell cycle progression and the initiation of apoptotic programs. Consistent with the data from the cell cycle analysis, Annexin/FITC V assays on HT29 cells revealed a significant apoptosis induced by FOPF. Moreover, the observed abnormalities in the cell cycle in all three cell lines could be indicative of a general framework where treated cells remain viable but become metabolically inactive, leading to the cytostatic effect induced by FOPF. This condition was mirrored by the relevant antimigration activity assessed by the wound-healing assay in LoVo and HT29 cells. This inhibition of migration is crucial for metastasis prevention, as it hinders the ability of cells to spread and proliferate in distant sites. In this work we have demonstrated the chemopreventive properties of FOFP. In this fraction, the presence of compounds associated with cytotoxic activity on cancer cell lines was identified. For example, ricinoleic acid was recently demonstrated to be active against both HT29 [33,34] and Hep3B [35] cell lines. Similar antitumor effects were observed with γ-tocopherol in both in vitro and in vivo tumor models [34,36,37] and with γ-sitosterol [16], both of which are present at high levels in our FOFP. Similar results were obtained with gamma-tocopherol [34]. Further studies aimed at defining the cytotoxic potential of each metabolite and their mechanism of action will be presented in due course.

## 4. Materials and Methods

### 4.1. Materials

Solvents and additives were purchased from Sigma–Aldrich (Burlington, MA, USA). Farclo apricot seed oil was obtained by the cold press of seeds deriving from the production campaign of 2018 by Società Agricola Guidi (Roncofreddo, Italy). It was stored in dark conditions at −20 °C until extraction.

### 4.2. Instruments

MARSX press, CEM Corporation, Matthews, NC, USA. Thermo ISQ GC/MS system (ISQ-MS coupled to Thermo Trace 1300/1310 GC with an autosampler apparatus). MS spectrometer Thermo Scientific (Milan, Italy) ISQ Single Quad system using Trace GC as a chromatographic device (Milan, Italy). Heidolph Laborota 4000 instrument (Heidolph Instruments GmbH & Co., Schwabach, Germany).

### 4.3. Extraction and Fractionation

Five mL of crude oil was added with five mL of ethanol and subjected to microwave heating (1 min ramping, 5 min hold time, maximum pressure 250 psi, maximum potency 80 W, temperature 60 °C). This procedure was repeated three times renewing the solvent, obtaining 313 mg of ethanolic extract (Farclò_EtOH_), which was further fractionated by liquid/liquid extraction (methanol/hexane). The methanolic fraction evaporated at a reduced pressure, furnishing a white solid, and a Farclo-purified fraction (FOPF) (9 mg). The hexane fraction also evaporated, furnishing an oil (297 mg).

### 4.4. Gas Chromatography

Column: Restek Rxi: 30 m, 0.25 mm i.d., 0.25 mm internal thickness.

Operative conditions: carrier gas = He, T-inj = 280 °C, T-TransferLine = 300 °C, ion source = 350 °C. Carrier flow = 1.0 mL/min, injection with split/splitless injector device with a 40:1 split ratio. Injection volume = 1 mL. The rate will be as follows: 60 °C (1 min), 10 °C/min (300 °C). and 300 °C (10 min).

Analysis of instrumental original datasets: NIST Mass Spectra Search Program^1f^ (version 2.4, build 25 March 2020) and AMDIS v 2.73 (25 April 2017 release), build 149.31.

### 4.5. In Vitro Experiments

#### 4.5.1. Cell Lines Culture

Colon adenocarcinoma cell lines (LoVo and HT29) and HCC cell line (Hep3B) were obtained from the American Type Culture Collection (ATCC, Manassas, UA, USA). Cell lines were maintained as a monolayer at 37 °C and 5% CO_2_. LoVo and Hep3B cells were cultured in Ham’s F-12K (Kaighn’s) Medium (ATCC^®^, Manassas, UA, USA) supplemented with 10% Foetal Bovine Serum (FBS) (Euroclone^®^, Milan, Italy) and EMEM (ATCC^®^, Manassas, UA, USA) with 10% FBS, respectively. HT29 cells were cultured in Roswell Park Memorial Institute RPMI (Euroclone^®^, Milan, Italy) supplemented with 10% FBS. For the treatment with FOPF cell media were supplemented with 1% Penicillin-Streptomicin (Gibco^®^, Thermo Fisher Scientific, Waltham, NA, USA) and 2% Amphotericin B (Euroclone^®^, Milan, Italy).

#### 4.5.2. Cell Growth Assay

Cell growth was determined with CellTiter96AQueous One Solution (Promega^®^, Milan, Italy). LoVo, HT29, and Hep3B cells were seeded in a 96-well plate at a density of 2.500 cells per well. Twenty-four hours after seeding, cells were treated with a medium containing FOPF at 0.001 mg/mL, 0.01 mg/mL, and 0.1 mg/mL. Every 24 h, the culture medium was replaced with fresh medium containing the same concentrations of FOPF. This process was repeated for 72 h, following a daily administration schedule. The absorbance of the 96-well plate was analyzed with BioTek Synergy^TM^ H1 microplate reader at 24, 48, and 72 h at 490 nm. All experiments were performed in triplicate. Growth inhibition and cytocidal effect of FOPF were calculated according to Monk’s formula: [(OD_treated_ − OD_zero_)/(OD_control_ − OD_zero_)] × 100, when OD_treated_ is ≥ to OD_zero_. If OD_treated_ is below OD_zero_, cell killing has occurred, while cell growth inhibition takes place if OD_treated_ is lower than OD_control_ but higher than OD_zero_. The OD_zero_ represents the cell number at the moment of FOPF addition, the OD_control_ reflects the cell number in untreated wells, and the OD_treated_ reflects the cell number in treated wells on the day of the assay [38].

#### 4.5.3. Wound-Healing Assay

Cells were seeded into a culture-insert two-well in μ-Dish (Ibidi^®^, Gräfelfing, Germany) at a density of 5.0 × 10^5^ per well. After 24 h the insert was removed and the medium was replaced with a fresh medium containing FOPF (0.001 mg/mL, 0.01 mg/mL, and 0.1 mg/mL). This protocol was repeated for the following 2 days (48 h and 72 h). Cell migration trend was monitored at 0, 24, 48, and 72 h after the treatment through EVOS microscope and calculated analyzing the cell-free area, using the wound-healing size tool by ImageJ software. All experiments were performed in triplicate.

#### 4.5.4. Annexin V/FITC

Cells were seeded in 25 cm^2^ flasks at a proper density. Twenty-four hours after seeding, cells were treated with FOPF 0.1 mg/mL. After 24 h, cell medium was replaced with a fresh medium containing FOPF 0.1 mg/mL. Every 24 h, the culture medium was replaced with a fresh medium containing the same concentrations of FOPF. After 72 h from the first treatment, cells were detached and processed with Thermo Fisher Annexin V/FITC Kit^®^ (Thermo Fisher Scientific, Waltham, NA, USA) and read with cytofluorimeter Attune NXT^®^ (Thermo Fisher Scientific, Waltham, NA, USA). All experiments were performed at least in quadruplicate.

#### 4.5.5. Cell Cycle Analysis

Cells were seeded in 25 cm^2^ flasks at a proper density. Twenty-four hours after seeding, cells were treated with FOPF 0.1 mg/mL. After 24 h cell medium was replaced with a fresh medium containing FOPF 0.1 mg/mL and the same treatment was performed after 24 h, reaching the daily administration scheme. After 72 h from the first treatment, cells were detached, fixed with 70% ethanol and stored at −20 °C for at least 24 h. After the fixation, cells were rinsed in PBS and stained in solution containing NP40, RNAse, and Propidium Iodide (PI). Cell cycle analysis was assessed through flow cytometry (Attune NXT^®^). All experiments were performed at least in triplicate.

#### 4.5.6. qRT-PCR

Cells were seeded in a 25 cm^2^ flask at a density of 210,000 cells. Twenty-four hours after seeding, the medium was replaced with a fresh medium containing FOPF 0.2 mg/mL. After 72 h of treatment, cells were detached and stored in Trizol^®^. After RNA extraction and quantification through Nanodrop™, retrotranscription was performed using iScript™ cDNA Synthesis Kit. PPAR-γ expression was measured by RT-PCR using HPRT and GAPDH as housekeeping, through 7500 Applied Biosystem^®^ (Thermo Fisher Scientific, Waltham, NA, USA). All experiments were performed at least in triplicate.

### 4.6. Statistical Analysis

Quantifiable data were derived from at least three independent experiments and were expressed as mean percentage relative to the unexposed control ± standard error of the mean (SEM). The statistical analysis was carried out using Graph Pad Prism 8 software and applying the Student’s *t*-test for 2-group comparisons.

## 5. Conclusions

This study presents FOPF, a novel purified fraction from Farclo apricot kernel oil, characterized by significant concentrations of bioactive compounds including ricinoleic acid and gamma-tocopherol. Our investigation demonstrated FOPF’s anticancer potential through multiple mechanisms: cytostatic effects across different cancer cell lines (LoVo, HT29, and Hep3B), significant inhibition of cancer cell migration, and cell cycle modulation. The observed effects appear to be mediated through PPAR-γ expression inhibition, suggesting a specific molecular pathway for FOPF’s anticancer activity. Of particular interest is FOPF’s differential impact on various cancer cell lines, showing both cytostatic effects and apoptosis induction depending on the cellular context. The ability of FOPF to inhibit both colorectal cancer and HCC cell line growth, combined with its natural origin and defined chemical composition, positions it as a promising candidate for nutraceutical development. While these in vitro results are encouraging, future research should focus on in vivo studies to validate FOPF’s chemopreventive potential and establish optimal dosing regimens. Additionally, investigations into the individual contributions of FOPF’s constituent metabolites will be crucial for understanding its mechanism of action.

These findings lay the groundwork for developing a targeted dietary supplement for individuals at high risk of colorectal carcinoma or HCC, particularly those with a history of these cancers. Such a preventive approach could represent a significant advancement in cancer chemoprevention strategies, offering a natural and potentially effective option for cancer risk reduction.

## Figures and Tables

**Figure 1 ijms-26-09237-f001:**
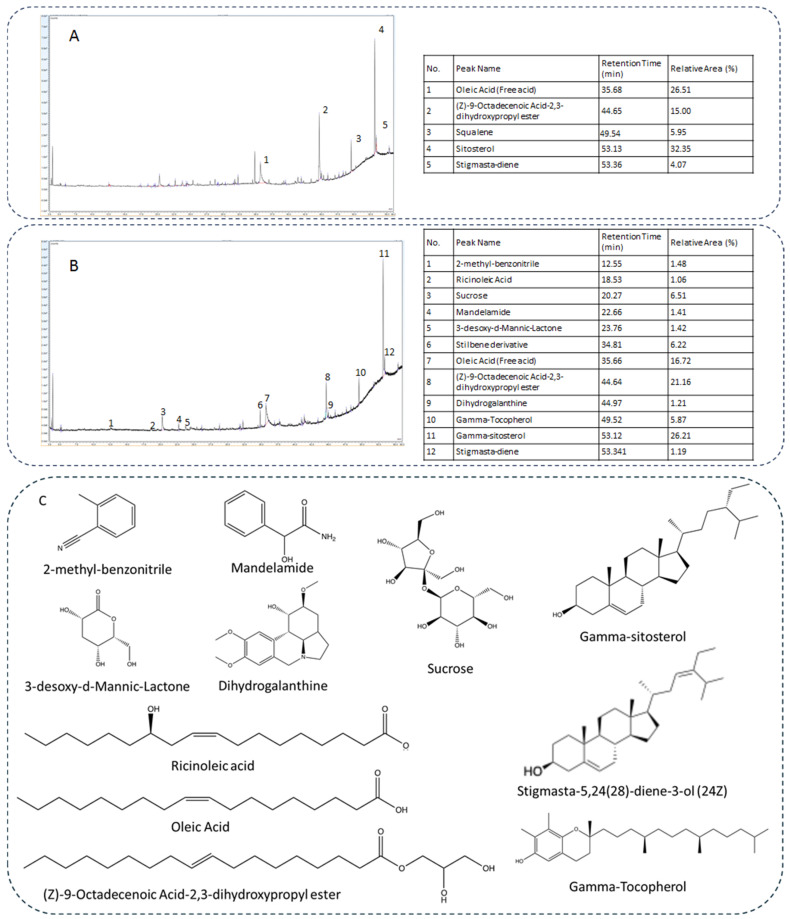
Chromatographic analysis. Chromatographic profile and relative metabolite abundance (expressed as relative area %) of the first L/L fraction (**A**), FOPF (**B**), and structures of metabolites identified in FOPF (**C**). Library match probability (match percentage) was determined using the relative abundance of ions (son ions), starting from the molecular ion.

**Figure 2 ijms-26-09237-f002:**
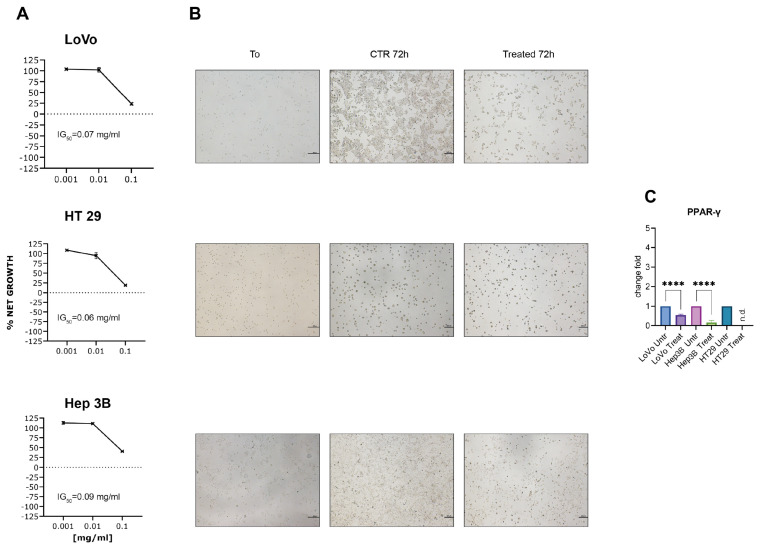
The effect of FOPF on cell growth and PPAR-γ expression. (**A**) Cell growth inhibition activity of FOPF after 72 h with repeated administration. FOPF was used at concentrations of 0.001 mg/mL, 0.01 mg/mL, and 0.1 mg/mL. Each point indicates the mean of at least three experiments. (**B**) Representative images of the three cell lines at different time points compared to the control. Images were taken at day 0 (T0) and after 72 h with repeated administration. (**C**) PPAR-γ expression in LoVo, Hep3B, and HT29 cells after FOPF treatment. Experiments were performed at least in quadruplicate. Data are expressed as mean ± SEM. *T*-test analysis was performed (****) *p* < 0.0001.

**Figure 3 ijms-26-09237-f003:**
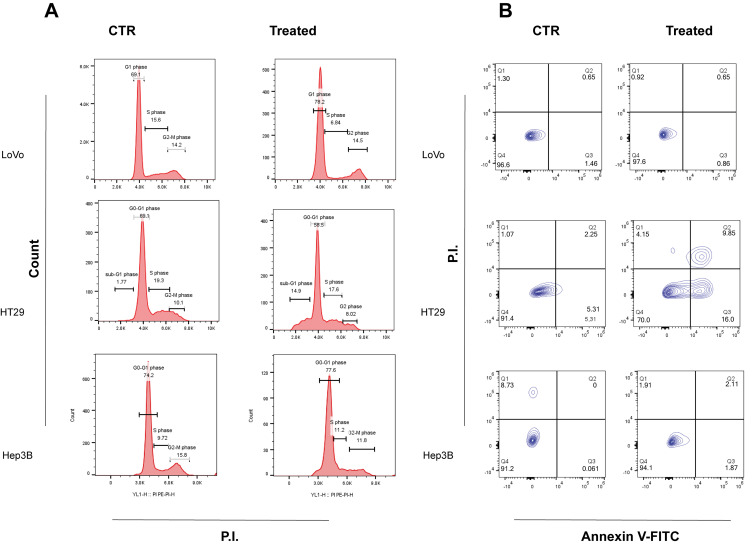
Cell cycle impairment induced by FOPF. (**A**) Effect of FOPF on LoVo, Hep3Band HT29 cell cycle progression after 72 h with repeated administration of 0.1 mg/mL dose. (**B**) Cell death after 72 h as evinced by Annexin V/FITC Assay. PI = Propidium Iodide. Images are representative of at least four experiments.

**Figure 4 ijms-26-09237-f004:**
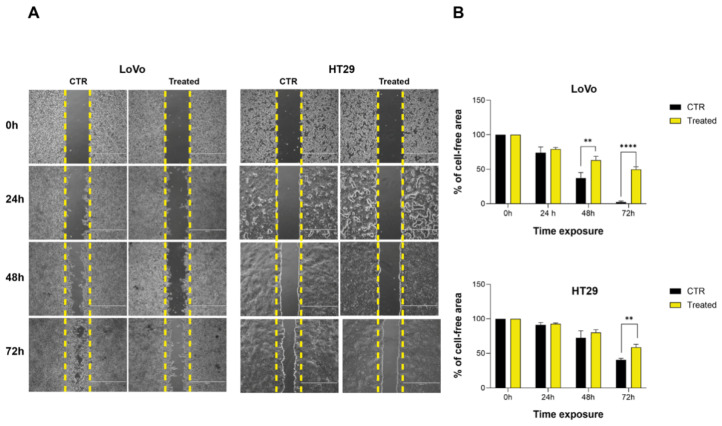
FOPF reduces cell migration. (**A**) Cell migration blockage induced after 72 h with repeated administration of FOPF 0.1 mg/mL in LoVo and 0.001 mg/mL in HT29. Images of control and treated cells were taken at 0 h, 24 h, 48 h, and 72 h after treatment. (**B**) Gap area analysis was performed using wound-healing plugin ImageJ and the resulting percentage was compared to the control. Data are expressed as mean ± SEM. *T*-test analysis was performed (**) *p* < 0.01, (****) *p* < 0.0001.

**Figure 5 ijms-26-09237-f005:**
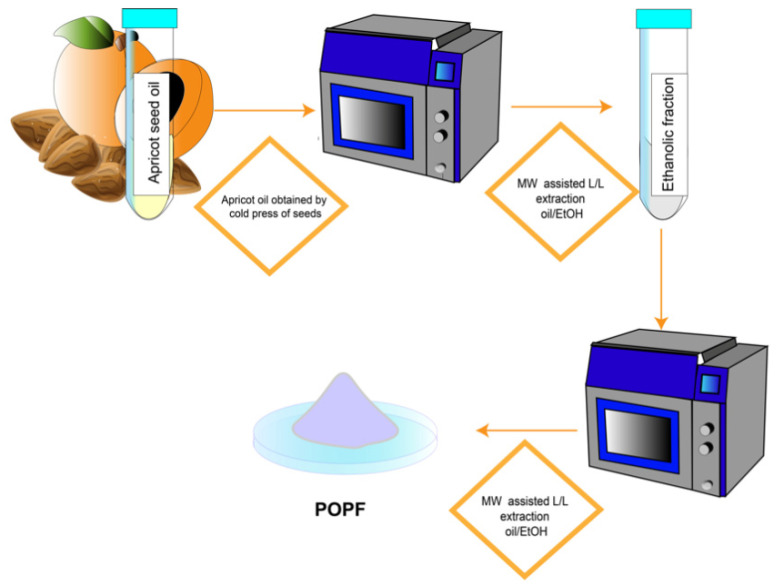
Schematic representation of the fractionation process to obtain FOPF.

## Data Availability

The original contributions presented in this study are included in the article. Further inquiries can be directed to the corresponding author

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
