# Peer review of "Preparation of a Micronutrient-Enriched Apricot Kernel Oil and Assessment of In Vitro Chemopreventive Propertiesâ€"

_ijms, 2025, doi:10.3390/ijms26189237_

Round 1
Reviewer 1 Report
Comments and Suggestions for Authors
The manuscript is very well structured and addresses a highly relevant topic within the field of bioactives and nutraceuticals. The focus on apricot kernel, a by-product, is particularly valuable as it combines agro-food waste valorization with potential therapeutic applications. The study presents solid results, with consistent antiproliferative activity data and an interesting emphasis on the role of PPAR-γ. The writing is clear, and the results are presented in a way that makes the paper both engaging and easy to follow. Overall, this is a valuable contribution to the literature, and I consider the work highly publishable in its current form, with only minor adjustments if the authors wish to further strengthen the discussion.
Author Response
Manuscript ID: ijms-3841960
Manuscript title: Preparation of a micronutrient-enriched apricot kernel oil and assessment of in vitro chemopreventive properties
Dear Editor,
please find attached a revised version of the manuscript entitled “Preparation of a micronutrient-enriched apricot kernel oil and assessment of in vitro chemopreventive properties”
The reviewers’ comments were highly enlightening and helped us to improve the manuscript’s quality. Revisions are shown in red font. The authors hope that the Reviewers and Editor will be satisfied with the further amendments which we have made after taking on board the feedback.
Reviewer 1
Comments:
The manuscript is very well structured and addresses a highly relevant topic within the field of bioactives and nutraceuticals. The focus on apricot kernel, a by-product, is particularly valuable as it combines agro-food waste valorization with potential therapeutic applications. The study presents solid results, with consistent antiproliferative activity data and an interesting emphasis on the role of PPAR-γ. The writing is clear, and the results are presented in a way that makes the paper both engaging and easy to follow. Overall, this is a valuable contribution to the literature, and I consider the work highly publishable in its current form, with only minor adjustments if the authors wish to further strengthen the discussion.
Response: The authors thank the reviewer for the positive feedback. In agreement with reviewer’s request we made adjustments in the discussion. In particular, lines 315-317 were added along with reference number 16-34-36-37.
Reviewer 2 Report
Comments and Suggestions for Authors
The authors presented a well-structured study “Preparation of a micronutrient-enriched apricot kernel oil and assessment of in vitro chemopreventive properties”. The findings in the manuscript are interesting to the field; however, at present version, several concerns are raised.
Recommendations for Revision and Improvement
1) The Introduction requires more depth and should present a stronger rationale for the study.
2) It is also recommended to cite additional relevant references to support the claims and provide a comprehensive background. These potential applications would strengthen the context.
To line 88-89 “Moreover, microwave induction can cause the degradation of amygdalin, drastically reducing its content” and also for the Introduction (Lines 52-54)
The Reference
M.A. Sheikh, M. Ubaid, N. Ahmed, M.K. Gul “Effects of controlled microwave heat treatment on the compositional attributes, antioxidant potential, and anti-nutritional components of apricot kernel flour. J. Food Meas. Charact. – 2025. – V. 19. – P. 1859-1873. doi: 10.1007/s11694-024-03079-4
would probably be helpful.
3) Please add an explanation to the manuscript text justifying the choice of the method for obtaining the purified fraction (FOPF) for better reader understanding .
4). Figure 1. Chromatographic profile and relative metabolite abundance:
Due to the authors used GC-MS to identify of active compounds presented in FOPF by comparing the m/z to standard library. To make it trustable, please consider to provide the observed m/z, calculated m/z and differences in ppm unit of identified compounds and found MS spectra in three more columns of Tables in Fig.1. Because If the differences were too high such as more than 5-20 ppm how the authors predict and identify those compounds.
5). Line 114 “The regiochemistry features of double bond, for gamma-sitosterol in particular, were determined with an extensive analysis of fragments obtained in ion-source of EI instrument”
Better was: “Double bond arrangement …”
6) Figure 1. The structure of stigmastadiene must be corrected. Is it stigmasta-3,5-diene? (Line 103)
7) Part 4.3 Extraction and fractionation. Lines 291-294.
“From 313 mg of ethanolic extract: Farclo purified fraction ( (9 mg) and hexane fraction (397 mg)”.
The data must be corrected.
By addressing these points, the study will achieve a higher level of clarity, further supporting its publication.

Author Response
Manuscript ID: ijms-3841960
Manuscript title: Preparation of a micronutrient-enriched apricot kernel oil and assessment of in vitro chemopreventive properties
Dear Editor,
please find attached a revised version of the manuscript entitled “Preparation of a micronutrient-enriched apricot kernel oil and assessment of in vitro chemopreventive properties”
The reviewers’ comments were highly enlightening and helped us to improve the manuscript’s quality. Revisions are shown in red font. The authors hope that the Reviewers and Editor will be satisfied with the further amendments which we have made after taking on board the feedback.
Reviewer 2
Comments:
The authors presented a well-structured study “Preparation of a micronutrient-enriched apricot kernel oil and assessment of in vitro chemopreventive properties”. The findings in the manuscript are interesting to the field; however, at present version, several concerns are raised.
Recommendations for Revision and Improvement
1) The Introduction requires more depth and should present a stronger rationale for the study.
Response: To fulfil with the reviewer’s comment we added more information in the introduction.
Lines 57-68 were added along with the relative references (7-8). In addition, from line 79 to 95, a stronger rationale for the study was included with further references (15-16-17-18).
2) It is also recommended to cite additional relevant references to support the claims and provide a comprehensive background. These potential applications would strengthen the context.
To line 88-89 “Moreover, microwave induction can cause the degradation of amygdalin, drastically reducing its content” and also for the Introduction (Lines 52-54)
The Reference
M.A. Sheikh, M. Ubaid, N. Ahmed, M.K. Gul “Effects of controlled microwave heat treatment on the compositional attributes, antioxidant potential, and anti-nutritional components of apricot kernel flour. J. Food Meas. Charact. – 2025. – V. 19. – P. 1859-1873. doi: 10.1007/s11694-024-03079-4
would probably be helpful.
Response: The authors appreciate the reviewer's feedback and additional relevant references
have been included to strengthen the context. Particularly, reference number 7-8-15-16-17-18 were incorporated. Furthermore, the reference suggested by the reviewer was included as ref n° 6 in lines 54 and 111.
3) Please add an explanation to the manuscript text justifying the choice of the method for obtaining the purified fraction (FOPF) for better reader understanding.
Response: We thank the reviewer for the suggestion. We improved this aspect at the beginning of the results section (line 106-110).
4) Figure 1. Chromatographic profile and relative metabolite abundance:
Due to the authors used GC-MS to identify of active compounds presented in FOPF by comparing the m/z to standard library. To make it trustable, please consider to provide the observed m/z, calculated m/z and differences in ppm unit of identified compounds and found MS spectra in three more columns of Tables in Fig.1. Because If the differences were too high such as more than 5-20 ppm how the authors predict and identify those compounds.
Response: Due to instrumental limitations, the library match probability (match percentage) was determined using the relative abundance of ions (son ions), starting from the molecular ion. The reviewer requested to evaluate the difference in ppm between the calculated and observed m/z values; however, this is not applicable with this instrument, as it does not have high-resolution mass spectrometry (HRMS) capabilities.
5) Line 114 “The regiochemistry features of double bond, for gamma-sitosterol in particular, were determined with an extensive analysis of fragments obtained in ion-source of EI instrument”
Better was: “Double bond arrangement …”
Response: The authors corrected the sentence as suggested by the reviewer
6) Figure 1. The structure of stigmastadiene must be corrected. Is it stigmasta-3,5-diene? (Line 103)
Response: The panel C of Figure 1 was corrected
7) Part 4.3 Extraction and fractionation. Lines 291-294.
“From 313 mg of ethanolic extract: Farclo purified fraction ( (9 mg) and hexane fraction (397 mg)”.
The data must be corrected.
Response: The authors apologize for the mistake, we corrected the data
By addressing these points, the study will achieve a higher level of clarity, further supporting its publication.